# *In Vitro* Digestibility, Biological Activity, and Physicochemical Characterization of Proteins Extracted from Conventionally and Organically Cultivated Hempseed (*Cannabis sativa* L.)

**DOI:** 10.3390/molecules28030915

**Published:** 2023-01-17

**Authors:** Mohammadreza Khalesi, Luthando Gcaza, Richard J. FitzGerald

**Affiliations:** Department of Biological Sciences, University of Limerick, Limerick, Ireland

**Keywords:** hempseed protein, organic cultivation, digestibility, antioxidant activity, lipase inhibitory activity

## Abstract

The proteins from two conventionally (CC1 and CC2) and one organically cultivated (OC) hempseed samples were extracted (by alkaline solubilization followed by isoelectric precipitation) and compared in terms of their physicochemical, digestibility and *in vitro* bioactivity properties. The OC hempseed had higher total protein and lower nonprotein nitrogen content. Protein extracts showed bimodal particle size distributions, with OC showing the smallest and CC1 the largest mean particle diameter (d(0.5)), i.e., 89.0 and 120.0 µm, respectively. Chromatographic analysis showed similar protein profiles for all three protein extracts. The protein extracts were subjected to *in vitro* simulated gastrointestinal digestion (SGID). Degree of hydrolysis (DH) measurement showed that the highest extent of digestion upon SGID was associated with CC1 (11.0 ± 1.5%), which also had the lowest *in vitro* antioxidant activity. Only the OC and OC digested samples had lipase inhibitory activity. The results indicate that the cultivation method impacted the composition, physicochemical, digestibility, and biofunctional properties of hempseed proteins.

## 1. Introduction

The demand for high-quality plant protein sources is increasing due to developments related to food sustainability and to consumer awareness on the potential impact of animal-sourced proteins on climate change, economic cost and health. Hemp (*Cannabis sativa* L.) is an annual herbaceous plant that has been widely used due to its industrial [1], ornamental [2] and pharmaceutical [3] applications. Around 50,000 ha hemp was cultivated in the EU in 2018 (https://eiha.org/wp-content/uploads/2020/06/2018-Hemp-agri-report.pdf (accessed on 7 November 2020)). It is mainly used for the production of edible oil (due to its high content of polyunsaturated fatty acids, especially ω-3 and ω-6 fatty acids), for biofuel, and in the textile and construction industries.

Hempseed contains approximately 30% (*w*/*w*) oil, 10–15% (*w*/*w*) insoluble fiber and 25% (*w*/*w*) protein. The presence of such high amounts of protein in hempseed highlights its potential for the development of plant-based protein enriched ingredients. Hempseed protein principally consists of albumin (~30%) and the legumin termed edestin (~70%) and contains high levels of the essential amino acids (EAA) [4,5]. The high level of arginine is particularly desirable for health-conscious consumers who ingest hempseed protein as a dietary component for maintenance of cardiovascular health [6]. Furthermore, the low level of protease inhibitors in hempseed is believed to contribute to the enhanced digestibility of hempseed proteins. Protein digestibility (*in vitro*) and protein digestibility corrected amino acid score (PDCAAS) values of up to 92% and 0.66, respectively, have been reported for hempseed protein [6]. Furthermore, hempseed proteins have significant potential as ingredients in the formulation of various food products, especially oil-in-water emulsions [7].

The hempseed proteome contains peptide sequences with potential beneficial health effects on the gastrointestinal, cardiovascular, immune and nervous systems. The *in vitro* 2,2-diphenyl-1-picrylhydrazyl radical (DPPH^●^) scavenging activity of hempseed protein hydrolysates at a concentration of 1 mg/mL was determined. The 1 kDa permeate fraction had a higher (24.2%) antioxidant activity compared to the 5–10 kDa fraction (18.7%) [5]. Using computational and database (BIOPEP) analysis, Aiello et al., [8] identified peptide SHLNWVCIFLGFHSFGLYI as potentially the most potent bioactive sequence, e.g., having antioxidant, dipeptidyl peptidase IV (DPP-IV)-inhibitory, and angiotensin-converting enzyme inhibitor (ACE)-inhibitory activity among 22 bioactive sequences within hempseed protein. It was indicated that that different di- and tripeptide fragments from the above sequence (compared to other identified sequences) potentially possessed different bioactivities, e.g., HL and LY have antioxidant activity.

Hemp grows well in a variety of climates and soil types and has an excellent ability to grow without fungicides, herbicides and pesticides [6]. Hemp can also be readily grown organically, and hence it is compatible with the dietary choices of people who prefer chemical-free food/food ingredients. However, to our knowledge, there is no literature on the impact of conventional versus organic cultivation of hemp on its protein quality. The hypothesis is that the method of hemp cultivation may have an impact on the protein components and their digestibility. Therefore, the objectives of this study were to extract proteins from conventionally and organically cultivated hemp and to characterize the extracted hempseed proteins in terms of their physicochemical, digestibility and *in vitro* bioactive properties.

## 2. Results and Discussion

### 2.1. Proximate Analysis of Differently Cultivated Hempseeds and Commercial Hemp Protein Isolate (HPI)

The composition of the hempseed samples and the commercial HPI is summarized in Table 1. Kjeldhal protein analysis showed that the protein content (23.73 ± 1.38%) of the OC sample was numerically higher (but not significantly different (*p* > 0.05)) compared to CC1 (21.93 ± 0.14%) and CC2 (21.18 ± 0.39%). The total protein content in the commercial HPI was 41.20 ± 1.25%. The protein content in hempseeds from 10 Canadian cultivars ranged between 22% and 26% [9]. Shen et al. [10] reported that dehulling hempseeds increased the protein content to ~27.5%. In a study by Tang, Ten, Wang, and Yang [11], the protein content in HPI was 50.2%, while in a study by Teh and Birch [12], this was 33.45%. The extraction method and downstream processing during protein extraction (purification and drying), the pH, the temperature and ionic strength employed for extraction of the proteins, the inclusion of a defatting step, seasonal variables, the hempseed variety, the location of cultivation, and the environmental conditions during growth may all impact the protein content and functional properties of hempseed protein extracts and thereby may impact their utilization in various composite food systems [13]. In addition, a nitrogen-to-protein conversion factor of 6.25 has been used in some reports, which gives an overestimation for the protein content. The nitrogen-to-protein conversion factor considered herein to calculate the hempseed protein content was 5.32 [14].

The NPN content in CC1 (3.28 ± 0.07%) and HPI (3.16 ± 0.14%) was significantly (*p* < 0.05) higher than in OC (2.61 ± 0.19%) and CC2 (2.84 ± 0.07%). Potin, Lubbers, Husson, and Saurel [15] previously reported an NPN of 4.3% in hempseed press cake.

The moisture content (7.30 ± 0.24%) of the OC hempseed was significantly (*p* < 0.05) higher than that of the other samples, i.e., 6.37 ± 0.21 and 6.37 ± 26% for CC1 and CC2, respectively. All the hempseed samples had a higher moisture content than the HPI (4.45 ± 0.05%) sample. The moisture values for the hempseed samples (range 6.37–7.30%) were comparable with the reported average value (6.2%) in hempseed from 10 Canadian cultivars [9]. Tang, Ten, Wang, and Yang [11] reported a moisture content for HPI of 6.7%, while the value reported by Teh and Birch [12] was 7.14%.

The ash content for all the hempseed protein extracts was similar (ranging from 4.81% to 5.10%) and was significantly (*p* < 0.05) lower than that of the HPI sample (8.38 ± 0.16%). The ash content in all three cultivated hempseed samples herein was similar to the mean values of ash content (5.2%) in hempseed from 10 Canadian cultivars [9]. The study by Tang, Ten, Wang, and Yang [11] reported an ash value in HPI of 3.2%, while in a study by Teh and Birch [12], this was 5.93%.

The lipid content in CC2 (31.32 ± 1.56%) was higher (*p* < 0.05) than for the two other samples (i.e., 24.71 ± 1.45% and 26.98 ± 0.72% for OC and CC1, respectively). All three hempseed samples used herein had significantly (*p* < 0.05) higher lipid content than HPI (10.13 ± 0.36%). The lipid content of the CC2 sample was similar to that previously reported for the Fasamo cultivar (30.5 ± 0.7%) grown in Canada [16]. The lipid content in the OC and CC1 samples were similar to the mean value (27.4%) in hempseed from 10 Canadian cultivars [9].

The mean values for the moisture, lipid, ash and protein content of 11 hempseed samples cultivated in different regions from three Canadian cultivars was reported to be 5.9%, 30.4%, 4.8%, and 24.0%, respectively [17]. However, the protein content in Canadian hempseed samples was calculated using a nitrogen-to-protein conversion factor of 6.25, as opposed to a value of 5.32 for the calculation herein. Considering a protein conversion factor similar to that used herein (5.32), the mean value for the protein content in the above Canadian samples was estimated to be 20.4%. Previously, the moisture, lipid, ash and protein contents in a Finola hempseed variety were reported to be 6.5%, 35.5%, 5.6, and 24.8%, respectively [4], while the moisture, lipid, ash and protein in a hempseed press cake were reported to be 5.3%, 30.2%, 5.2%, and 22.5%, respectively [15].

The dietary fiber content in OC (23.0 ± 1.3%) was significantly lower than CC1 (26.8 ± 0.6%) and CC2 (26.5 ± 0.6%). The dietary fiber of Finola variety hempseed was previously reported to be 27.6% [4]. The dietary fiber content of HPI was 18%.

Overall, the OC had the highest moisture (*p* < 0.05) and the lowest (*p* < 0.05) dietary fiber content. CC1 had the highest (*p* < 0.05) NPN and CC2 the highest (*p* < 0.05) lipid content. No significant difference (*p* > 0.05) between the protein content of the OC and CC samples was observed.

### 2.2. Solubility of the Commercial HPI Sample

Solubility is one of the most important properties of proteins, since it can affect many functional properties, e.g., the surface activity, rheological and hydrodynamic properties. The solubility of proteins is generally at its lowest value at the isoelectric point, where there is no net charge and therefore no repulsion between proteins. The solubility of proteins is thus strongly dependent on pH. The pH of maximum solubility is used for the extraction of proteins from different plant sources.

Therefore, the solubility of the commercial HPI at different pH (3–12) was determined in order to select a suitable pH for protein extraction from OC, CC1 and CC2. The results showed that protein solubility at alkaline pH was higher than under acidic conditions, with the highest (*p* < 0.05) solubility at pH 12 being equal to 82.2 ± 4.7% herein. The results showed that the solubility of the commercial HPI under neutral pH conditions was low. The protein solubility of HPI was minimum at pH 3–6, while it increased gradually above pH 6 (Figure 1). These data show that the commercial HPI was a typical alkali soluble protein.

The results herein are in agreement with the results obtained by Malomo and Aluko [7] reporting hempseed meal protein solubility in the range of 5–20% when suspended between pH 3 and 9. Tang et al. [11] reported a protein solubility > 90% when the hempseed protein was adjusted to pH ≥ 8. The results herein are also in agreement with the report by Potin et al. [15], who observed low solubility (<20%) of hempseed protein at pH < 9, while reaching solubility values of 30%, 50%, and 70% at pH 10, 11 and 12, respectively. The minimum solubility of defatted hempseed meal protein has been previously reported to be at pH 5–6 [18]. In general, hempseed protein exhibits low solubility at neutral and acidic pH in comparison to other plant proteins, such as soy protein [11,19]. This may be attributed to the aggregation of the acidic and basic subunits of edestin (globulin) at pH values ≤ 7 [20]. The underlying mechanism of solubilization of hempseed protein at alkaline pH may be related to dissociation of the edestin molecules. It has been reported that hempseed globulin has low solubility in the acidic pH range with values of approximately 20%, which increased to 50% at pH 8 and 9, while the solubility of albumin in hempseed ranged from 57% at pH 3 to 84% at pH 8, which is significantly superior to those of the globulin [7]. Some modifications, such as succinylation, acetylation, and enzymatic treatment, have been reported to improve the solubility of hempseed proteins [18].

The highest solubility for the HPI sample was obtained at pH 12. This value was therefore employed for the extraction and solubilization of the hempseed proteins from defatted OC, CC1 and CC2 samples followed by subsequently isoelectric precipitation at pH 4 (see Section 3.4).

### 2.3. Hempseed Extract Protein Profiles

The spray-dried CC2 protein extract powder contained significantly (*p* < 0.05) higher protein than the two other samples. The results showed that the spray-dried OC, CC1 and CC2 extracts contained 65.54 ± 1.24, 62.58 ± 1.61, and 70.03 ± 0.94% protein, respectively. The protein profiles of the spray-dried samples were analyzed using RP-UPLC (Figure 2). The protein profiles of all three samples were similar, showing dominant peaks on the chromatograms eluting at retention times of 25.5, 26.5 and 27.0 min. The elution of edestin at a retention time of ~30 min using a similar RP-HPLC gradient as herein was recently reported [21]. Therefore, the peaks appeared on the chromatogram herein at 25–27 min may correspond to the subunits of edestin. The area under the curve for the region associated with edestin for OC, CC1 and CC2 was 77.6%, 67.7%, and 73.4% (as a percentage of the total chromatogram area), respectively. The difference may depend on differences in the growth conditions. A previous study reported that hempseed proteins consist of ~65–75% edestin and 25–35% albumin [22].

### 2.4. Particle Size Distribution

The results of the particle size analysis of the aqueous reconstituted hempseed protein extracts showed that the smallest particles were associated with the OC sample while the largest was associated with the CC1 sample. The particle sizes for the CC2 sample were in between these two samples. Nevertheless, these values were lower than the mean particle diameter of commercially available HPI with 50% protein, which were recorded to be 200 µm (Cambridge Commodities Ltd., P19311 product specification). The uniformity of the protein extract samples was not significantly different (*p* > 0.05); however, the CC1 sample had the widest distribution of particle sizes (d(0.9) − d(0.1) = 590.0 µm) (Table 2). Bimodal particle size distributions were observed in all samples (Figure 3). This may be linked to the presence of two major proteins (i.e., albumin and edestin) showing different interaction patterns at pH 7 in aqueous suspension. Lower particle size and a narrower distribution of sizes are desirable for some applications (e.g., emulsification) of food proteins. The specific surface area (SSA) of the particles was not significantly different (*p* > 0.05).

### 2.5. Hempseed Protein Digestibility

#### 2.5.1. Degree of Hydrolysis (DH) of Hempseed Proteins after Simulated Gastrointestinal Digestion (SGID)

The DH of the digested OC, CC1 and CC2 samples subjected to the gastric and gastric followed by intestinal digestion was determined (Table 3). The DHs of the samples subjected to gastric digestion were not significantly (*p* > 0.05) different. The results showed that the highest (*p* < 0.05) DH obtained for the samples subjected to gastric followed by intestinal digestion was associated with the CC1 sample (11.0 ± 1.5%) followed by CC2 (8.7 ± 0.4%) and OC (8.1 ± 0.9%).

The incubation of HPI preheated at 75, 80 and 90 °C with trypsin for 10 min was previously reported to result in DHs of 2.3%, 4.5%, and 6.7%, respectively [16]. In another study, HPI was hydrolyzed using different conditions and different enzymes, i.e., pepsin (2%, 4 h), pepsin (4%, 4 h), Alcalase (1%, 4 h), Alcalase (2%, 4 h), papain (2%, 4 h), and pepsin + pancreatin (2%, 2 h + 4 h), resulting in the generation of hydrolysates with DH values of 5%, 7%, 18%, 26%, 17%, and 28%, respectively [23].

#### 2.5.2. SDS-PAGE Analysis of OC, CC1 and CC2 Proteins before and after SGID

The two main polypeptide chains, the 33 kDa acidic subunit and 18–20 kDa basic subunits of edestin, were observed on the gel of undigested hempseed proteins (Figure 4, lanes 2, 5 and 8). The reducing gel conditions used in this study separated the acidic subunit (with the highest intensity) and two basic subunits upon dissociation of edestin, especially for the lanes associated with undigested hempseed proteins (Figure 4, lanes 2, 5 and 8). Other minor polypeptides, such as those below 18 kDa (corresponding to albumin storage protein [11]), were also observed in all three samples. Among the samples tested herein, the proteins from the OC extract generally appeared to be better separated on the gel.

From the SDS-PAGE, the gastric digestibility of the OC protein extract (lane 3) appeared to be lower than that for the CC1 (lane 6) and CC2 (lane 9) protein extracts. This was in accordance with the DH results (Table 3). At the end of gastric digestion, the subunits of edestin and albumin were still visible in the OC sample (lane 2). The albumin subunits were not visible after gastric digestion of the CC1 (lane 6) and CC2 (lane 9) protein extracts, while the edestin subunits were still present. After SGID, some of the edestin subunits were still visible in the OC protein extracts (lane 4), while these were not observed for CC1 (lane 7) and CC2 (lane 10) protein extracts. The CC1 protein extract appeared to be digested more extensively compared to the CC2 and OC samples. The lower digestibility may be linked to the presence of higher levels of antinutritional factors (ANF). However, this warrants further study on the impact of cultivation on the level of ANF in the hempseed. Russo and Reggiani [24] reported that the ANF in hempseed is cultivar type- and geographic location-dependent according to their investigation on the level of ANFs (including phytic acid, condensed tannins, trypsin inhibitors, cyanogenic glycosides and saponins) among three Italian (Carmagnola, CS, Fibranova) and three French (Fedora 17, Felina 32, Ferimon) hempseed varieties. They showed that the highest ANF in all varieties was associated with phytic acid, which was in the range of 60–70 mg/g in hempseed meal [24]. This level of phytic acid in hempseed is much higher than in many other crops, e.g., soy with ~20 mg/g [25]. The average phytic acid content in hempseed meal from the cannabinoid-free hemp variety Helena was reported to be 22.5 mg/g [26]. Pojić et al. [26] showed that the phytic acid content was higher in fine fractions compared to the coarse fractions containing hull particles.

#### 2.5.3. Molecular Mass Distribution

Molecular mass distribution analysis showed that hempseed proteins having MW > 10 kDa were hydrolyzed to smaller components during SGID (Figure 5). The most components with MWs of 5–10 kDa was associated with the gastric digested OC sample. This was in accordance with the results of SDS-PAGE analysis (Figure 4). The gastric digested CC1 sample had the highest content of peptides: < 1 kDa. The gastrointestinal digested CC1 and CC2 samples had a lower content of peptides (MW < 1 kDa) than the gastric digests. This was not the case for the OC sample. Following SGID, the level of the peptides < 1 kDa was higher for the OC protein extract compared to CC1 and CC2 extracts. Peptides with MW < 1 kDa may be potentially bioactive. Previously, Nongonierma and FitzGerald [27] reported that the molecular mass distribution of an SGID digested commercial HPI sample had a distribution where components > 10 kDa, 5–10 kDa, 1–5 kDa and < 1 kDa were present at 1%, 5%, 30%, and 64%, respectively. This is similar to the mass distribution results of the post-SGID OC protein extract herein.

### 2.6. Antioxidant and Lipase Inhibitory Activity of Hempseed Proteins before and after SGID

The *in vitro* 2,2′-azino-bis(3-ethylbenzothiazoline-6-sulfonic acid) radical (ABTS^●^) scavenging activity analysis showed that at a concentration of 1 mg/mL, the hempseed protein extracts had low antioxidant activity. Subjection of the hempseed protein extracts to SGID increased their antioxidant activity. However, there was no significant difference (*p* > 0.05) between the antioxidant activity of the gastric digested and gastrointestinal digested sample from the different hempseed protein extracts. Among the intact proteins, the OC sample had the highest (*p* < 0.05) ABTS^●^ scavenging activity (16.9 ± 0.9%), while no significant difference (*p* > 0.05) was observed between the antioxidant activity of CC1 and CC2. Among the digested samples, the highest ABTS^●^ scavenging activity (86.9 ± 1.3%) was associated with the OC gastric digested samples, while the lowest (82.4 ± 0.6% at a concentration of 1 mg/mL on a protein basis) was associated with the gastric digested CC1 sample. Therefore, SGID treatment released peptides with ABTS^●^ scavenging activity.

Previously, hempseed protein hydrolysates at a concentration of 0.5 mg/mL were reported to have a DPPH^●^ scavenging activity of 20–60%, while their subsequently fractionated peptides with sequences corresponding to GSH, WVYY and PSLPA at 0.5 mg/mL had radical scavenging activity of 80%, 70%, and 60%, respectively [28]. The alkali soluble proteins/peptides of hempseed were reported to have higher DPPH^●^ scavenging activity compared to the acid soluble samples [29]. It should be noted that the presence of phenolic compounds in hempseed may contribute to the overall antioxidant activity of hempseed proteins/peptides. Previously, seasonal factors as well as cultivars have been reported to impact the content of phenolic compounds [30]. For example, it was reported that hempseeds from soils that did not undergo pre-seeding fertilization had higher phenolic contents than those that had [31].

Proteins and peptides with pancreatic lipase inhibitory activity are potentially promising compounds with antiobesity properties. The results herein showed that the intact and digested proteins derived from the two conventionally cultivated hempseeds (CC1 and CC2) had no lipase-inhibitory activity. In contrast, CC1 and CC2 protein extracts and also the digests of CC1 and CC2 at a protein concentration of 1 mg/mL had lipase-inducing activity up to 35–42%. Interestingly, the intact and digested proteins derived from the organically cultivated hempseed (OC) at a protein concentration of 1 mg/mL inhibited ~22% of the lipase activity. Differences between the OC and CC samples may be associated with compositional differences (Table 1). Limited studies are available on plant proteins/peptides with lipase-inhibitory activity. Jakubczyk, Szymanowska, Karaś, Złotek, and Kowalczyk [32] showed that the peptides obtained from millet grains inhibited the activity of pancreatic lipase. However, to our knowledge, there are no reports to date on the lipase-inhibitory/inducing activity of hempseed proteins/peptides. The results herein showed a lipase-inhibition effect from OC and its digests while it showed a lipase activation effect for the CC samples. Further investigations on the impact of residual lipid molecules on the lipase-inhibitory/inducing activity in protein extracts obtained from defatted hempseed are warranted.

## 3. Materials and Methods

### 3.1. Materials

Sodium hydroxide (NaOH), hydrochloric acid (HCl), trichloroacetic acid (TCA) and acetic acid were from Fisher Scientific (Dublin, Ireland). Kjeldahl tablets, sulfuric acid (>98%, H_2_SO_4_), boric acid, trifluoroacetic acid (TFA), Trizma^®^ base, dimethyl sulfoxide (DMSO), 2-mercaptoethanol, methanol, protein markers (6.5–200 kDa), Sudan III, 2,4,6-trinitrobenzenesulfonic acid (TNBS), 2,2′-azino-bis(3-ethylbenzothiazoline-6-sulfonic acid) (ABTS), porcine pepsin (2500 U/mg), porcine pancreatic lipase (50 mU/mL), 4-methylumbelliferyl oleate (4-MUO) and Orlistat^®^ were from Sigma-Aldrich (Dublin, Ireland). Hexane, MS-grade water and acetonitrile (ACN) were from Honeywell International Inc. (Dublin, Ireland). Comassie R, Laemmli buffer and Mini-Protean TGX 4–20% pre-cast polyacrylamide gels were from Bio-Rad Laboratories Inc. (Hercules, CA, USA). Sodium dodecyl sulfate (SDS) was from National Diagnostics (Atlanta, GA, USA). Corolase PP^®^ (4.4 U/mg) was from AB Enzymes GmbH (Darmstadt, Germany). Cellulose acetate filters were from Millipore (Carrigtwohill, Ireland) and polytetrafluoroethylene syringe filters were from VWR (Dublin, Ireland).

### 3.2. Sample Preparation

Two conventionally grown hempseed (CC1 and CC2) and one organically grown hempseed (OC) samples were provided by Hemptech Ireland Ltd., Wicklow, Ireland. The hemp *Cannabis sativa* L. (Finola)) was grown under license from the Irish Department of Health (license numbers 5C/134-1-2019, 5C/94-1-2019 and 5C/135-1-2019). The hemp was sown between 25 and 27 May 2019 and CC2 and CC1 were harvested between 11 and 12 October 2019, while the OC crop was harvested on 22 October 2019. The samples were cleaned by physical separation of the whole seeds from any other materials.

In addition, a commercial hempseed protein isolate (HPI, Pulsin Ltd., Gloucester, UK) was purchased from a local health-food store.

### 3.3. Proximate Analysis and Sample Characteristics

#### 3.3.1. Moisture, Ash, Dietary Fiber and Lipid Contents

During moisture content analysis, the hempseeds were firstly ground to fine particles using a KG49 Delonghi coffee grinder (Delonghi, Shanghai, China). Empty sample pans were weighed (M1) and an aliquot (1.5 g) of each hempseed sample and the commercial HPI was then added to each sample pan and the total weight was recorded (M2). Samples were dried (105 °C, 6 h) in a hotbox oven with fan size 2 (Gallenkamp Ltd., Loughborough, UK) and weighed again once dried (M3). The moisture content was calculated using Equation (1):(1)$$\mathrm{Moisture}\;\mathrm{content}=\frac{\left(M2-M1\right)-\left(M3-M1\right)}{\left(M2-M1\right)}\times 100\%$$

For ash content measurement, the oven-dried crucibles were cooled and weighed (W1) prior to adding 250 mg of each ground sample and the commercial HPI (n = 3). The total weight was obtained (W2). The samples were dried in a furnace (B180 Nobertherm Bremen, Germany) at 550 °C for 8.5 h and weighed again (W3) after cooling. The ash content of each powder was calculated using Equation (2):(2)$$\mathrm{Ash}\;\mathrm{content}=\frac{\left(W2-W1\right)-\left(W3-W1\right)}{W2-W1}\times 100\%$$

The lipid content of samples was determined according to AOAC (2000) [33]. An aliquot (1.5 g, n = 3) of each sample was added to preweighed round-bottomed flasks. Hexane (100 mL) was added to the flasks to extract total lipid from the sample during 6 h refluxing, followed by evaporation (70 °C, 10 min) using an RE100 rotary evaporator (Bibby Scientific, Stone, Staffordshire, UK). The flasks were reweighed to obtain the quantity of lipid extracted.

#### 3.3.2. Protein (PN) and Nonprotein Nitrogen (NPN) Determination

The level of nitrogen in each sample was determined (n = 3) using a KjelDigester K-446 (BUCHI Laboratories AG, Flawil, Switzerland) according to the Kjeldahl protein nitrogen determination method (IDF Provisional Standard 20B 1993). The protein was calculated using a nitrogen-to-protein conversion factor of 5.32 [14].

The NPN was determined using the TCA precipitation method [34]. Briefly, TCA (12% final concentration) was added to 0.8 g of ground sample (dissolved in dH_2_O 1:20 (*w*/*v*)) and the sample was allowed to stand for 180 min in order to precipitate the proteinaceous compounds. The sample was then centrifuged at 4 °C and 3500× *g* for 15 min using a 320R Hettich centrifuge (Tuttlingen, Germany). The nitrogen in the supernatant and the pellet were separately analyzed as described above. The nitrogen content in the supernatant indicated the NPN content.

### 3.4. Extraction of Hempseed Protein

The nitrogen solubility of the commercial HPI at different pH values was firstly assessed in order to select the optimum pH for extraction of the proteins from the hempseeds. For this, the HPI sample was suspended 1:20 (*w*/*v*) in dH_2_O. The HPI suspension was stirred for 10 min and the pH was then adjusted to pH 3, 4, 5, 6, 7, 8, 9, 10, 11 or 12 using HCl (0.5 M) or NaOH (0.5 M) followed by stirring for 30 min at room temperature. An aliquot of the suspension (7 mL) was centrifuged (3000× *g* at 4 °C for 15 min). The nitrogen content in the supernatant obtained was analyzed (n = 3) using the Kjeldahl procedure. The pH of the supernatant with the highest protein content (maximum solubility, pH 12) was subsequently selected for protein extraction from the hempseed samples.

Protein extraction from the hempseed samples was carried out according to the pH selected from the previous step that indicated the maximum solubility. First, the hempseed samples were de-oiled using a stainless steel laboratory automatic oil press (TTLIFE, Nanjing, Jiangsu, China). The de-oiled samples were subsequently ground using a coffee grinder. As described previously, the ground samples were then suspended in dH_2_O (1:20, (*w*/*v*)) and stirred at room temperature for 10 min. The pH was then adjusted to that of maximum protein solubility (pH 12). Samples were stirred for 30 min followed by centrifugation (3500× *g*, 4 °C, 15 min) using a 320R Hettich centrifuge (Tuttlingen, Germany). The pH of the supernatant was then adjusted to pH 4, i.e., the pH corresponding to the isoelectric point of hempseed protein [23] using 0.5 M HCl. The pellet collected after centrifugation (3500× *g*, 4 °C, 15 min) was resuspended in dH_2_O (1:10 (*w*/*v*)). The suspension was then dried using a B-290 mini spray dryer (Buchi, Flawil, Switzerland) at an inlet temperature of 150 °C and an outlet temperature of 85 °C. The protein content of each spray-dried powder sample was determined using the Kjeldahl method as outlined earlier.

### 3.5. Reverse-Phase Ultrahigh-Performance Liquid Chromatography (RP-UPLC)

The protein extracted from the OC and CC hempseeds was analyzed using an Acquity UPLC system (Waters, Dublin, Ireland). Mobile phase A was 0.1% (*v*/*v*) TFA and mobile phase B was 80% (*v*:*v*) ACN and 0.1% (*v*/*v*) TFA. The samples were reconstituted in dH_2_O to reach a concentration of 0.5% (*w*/*v*). The samples were filtered through 0.2 µm cellulose acetate filters. An aliquot of 10 μL was injected into the Acquity UPLC BEH C18, 130 Å column (2.1 mm × 50 mm × 1.7 mm) equipped with an Acquity BEH C18 (1.7 mm) vanguard precolumn. The flow rate was set on 0.3 mL/min over 51 min with the gradient program as previously detailed by Cermeño et al. [35].

### 3.6. Particle Size (PS) Analysis of Reconstituted Hempseed Protein Extracts

An aliquot (1 mL) of reconstituted (5% (*w*/*v*)) protein extract adjusted at pH 7.0 was pipetted into a Mastersizer 2000 (Malvern Instruments, Malvern, Worcestershire, England) equipped with a Hydro 2000S sample dispersion system (set at a stirring speed of 1000 rpm) interfaced with Mastersizer 2000 software (version 5.61; Malvern Instruments, Malvern, UK). The volume weighted mean (D[4, 3]), the surface weighed mean (D[3, 2]), the absolute deviation from the median (uniformity) as well as the average particle size (d(0.5)), the sizes of particles below which 10% and 90% of the sample lie (d(0.1) and d(0.9), respectively) and the specific surface area (SSA) were recorded. The particle refractive index and the continuous phase refractive index used were 1.52 and 1.33, respectively. The beam length was 2.35 mm and the minimum detection limit was 0.02 µm. Measurement integration time was 12,000 ms. The analysis was performed in triplicate with three measurements during each run.

### 3.7. Simulated Gastrointestinal Digestion (SGID) of Hempseed Protein Extracts

SGID was carried out on the hempseed protein samples according to Walsh et al., [36]. The samples (100 mL, 2% (*w*/*v*) protein) were incubated in a water bath (37 °C, 30 min). The pH was adjusted to pH 2 using 1 M HCl. Following addition of pepsin (2.5% (*w*/*v*) protein), gastric digestion took place for 90 min before the sample was heated to 90 °C for 20 min, deactivating the enzyme. The pH of the sample was then adjusted to pH 7.5 using 1 M NaOH. Intestinal digestion occurred with the addition of Corolase PP (1% (*w*/*v*) protein) and incubation for 150 min at 37 °C before deactivation. The pH values were kept constant throughout each digestion phase using a pH stat (Metrohm 902 Titrando pH-STAT, Herisau, Switzerland). The samples containing digested protein were then freeze-dried (Labconco benchtop freeze-dryer, Kansas City, MO, USA). Therefore, three samples were obtained from each hempseed protein extract: a sample prior to digestion, a sample that had undergone simulated gastric digestion and a sample that had undergone simulated gastric and intestinal digestion. These were subsequently characterized for their degree of hydrolysis (DH%), sodium dodecyl sulfate polyacrylamide gel electrophoresis (SDS-PAGE) and gel-permeation high-performance liquid chromatography (GP-HPLC) profiles, *in vitro* antioxidant activity, and lipase-inhibitory activity.

### 3.8. Degree of Hydrolysis (DH%)

The DH was determined according to the method described by Nongonierma et al., [37], where samples at 5% (*w*/*v*) protein were diluted in 1% (*w*/*v*) SDS to reach a concentration of 0.1% (*w*/*v*). Samples were heated for 30 min at 50 °C. The 5% (*w*/*v*) TNBS stock solution was diluted to 0.05% (*w*/*v*) using a 1:1 solution of dH_2_O and sodium phosphate buffer (0.2125 M, pH 8.2). An aliquot of 10 μL from each sample was then added to a 96-well microplate (Sarstetd, Dublin, Ireland) along with 160 μL of TNBS in each well. The absorbance was measured at 50 °C every 5 min during 1 h at λ = 350 nm using a microplate reader (BioTek Synergy HT reader, BioTek, Winooski, VT, USA). Leucine at various concentrations (0, 2, 5, 7, 14, 21, 28 and 56 mg/mL in 1% (*w*/*v*) SDS) was used as a standard.

### 3.9. Sodium Dodecyl Sulfate Polyacrylamide Gel Electrophoresis (SDS-PAGE) of Hempseed Protein Extracts

SDS-PAGE analysis was used to obtain the profiles of each hempseed protein sample. Samples (~4 mg/mL on a protein basis) were mixed 1:1 with a mixture of 2-mercaptoethanol and Laemmli buffer (1:19) in Eppendorf tubes. The tubes were shaken for 5 min at 200 rpm and incubated at 95 °C using a DX-100/DX-100R Dry Block Thermo-Shaker (IRIS/analytical, London, UK). An aliquot of 5 µL of a protein marker standard and 10 µL hempseed protein samples (~20 µg protein) was pipetted into each lane. The electrophoresis apparatus (Power PAC 1000, Bio-Rad, Hercules, CA, USA) was set at 150 V and 15–50 mA for 90 min.

### 3.10. Gel-Permeation High-Performance Liquid Chromatography (GP-HPLC) Analysis

Protein suspensions containing 0.25% (*w*/*v*) hempseed protein dissolved in mobile phase (30% ACN, 0.1% TFA) were prepared for GP-HPLC analysis. Samples were then filtered into Eppendorf tubes using 0.2 μm polytetrafluoroethylene syringe filters and 200 μL of each sample was transferred into HPLC sample vials. Separation was performed by isocratic elution on a TSK G2000 SW separating column (600 mm length × 7.5 mm ID) fitted with a TSKGEL SW guard column (75 mm length × 7.5 mm ID—Tosoh Bioscience, Tokyo, Japan) over 51 min. The absorbance of the eluent was monitored at 214 nm and separation was carried out at a flow rate of 1 mL/min.

### 3.11. ABTS^●^ Scavenging Activity Assay

Hempseed proteins and their digested samples (21 mg on a protein basis) were diluted to 1 mL with phosphate-buffered saline (PBS, pH 7.4) and centrifuged at 3500× *g* at 4 °C for 5 min. ABTS^●^ was diluted to an absorbance of ~0.7 at λ = 734 nm. An aliquot (200 μL) of this was added to 10 μL of samples. PBS was used as a blank. All samples including the blank were applied to a microplate and the absorbance recorded over 6 min, with shaking at 30 °C using a BioTek Synergy HT microplate reader. The scavenging by the samples of the ABTS^●^ was calculated according to Re et al. [38].

### 3.12. Lipase-Inhibitory Activity

The lipase-inhibitory activity assay was performed according to Amigo-Benavent et al. [39]. The assay buffer used was 13 mM Tris–HCl (pH 8.0) containing 150 mM NaCl and 1.3 mM CaCl_2_. Porcine pancreatic lipase was dissolved in assay buffer (1 mg/mL). The substrate was a solution of 4-methylumbelliferyl oleate (4-MUO) dissolved in DMSO to reach a concentration of 1 mM. Samples (1 mg/mL on a protein basis) were dissolved in DMSO. The assay was performed by addition of preheated (30 °C) lipase (25 µL, 50 mU/mL) to 50 µL of substrate, 25 µL test sample and 100 µL assay buffer in a microplate followed by shake-incubation at 30 °C for 30 min. Afterwards, the fluorescence was measured at excitation and emission wavelengths of 360 and 460 nm, respectively, using a BioTek Synergy HT microplate reader. The positive control was Orlistat^®^ which was prepared similarly to the test samples. The percentage of lipase inhibition of samples at a concentration of 1 mg/mL (on a protein basis) was reported.

### 3.13. Statistical Analysis

Data values are presented as the mean of three measurements ± standard deviation (SD). One-way analysis of variance (ANOVA) followed by the Tukey post hoc comparison test was carried out to test for significant differences using Minitab^®^ Release 15 for Windows, and *p* < 0.05 was considered statistically significant.

## 4. Conclusions

This study focused on the extraction and characterization of the proteins from two conventionally (CC1 and CC2) hempseed samples and one organically cultivated (OC) hempseed sample. The hempseeds contained 20–25% protein that consisted of edestin (65–75%) along with albumin molecules. Minor variations were observed in the protein profiles of the samples. The OC and CC1 hempseeds had the lowest NPN. The particles in suspension of the OC protein extract were more uniform and had a lower mean diameter than the CC samples. Subjection of the protein extracts to SGID resulted in differences in molecular mass distribution of the digested samples, with the OC samples showing lower MWs compared to CC protein extracts. In addition, the digested OC protein extracts had higher (*p* < 0.05) *in vitro* antioxidant and lipase inhibitory activity than CC samples. This study showed that the cultivation conditions had a considerable impact on the physicochemical and biofunctional properties of hempseed proteins. It should be mentioned that these are preliminary results from a limited number of samples and further studies would be required to better understand the impact of cultivation on the nutritional quality of hempseed proteins. More investigations on the possible applications of organically cultivated hempseed proteins as a sustainable alternative to animal-origin protein in food products are warranted.

## Figures and Tables

**Figure 1 molecules-28-00915-f001:**
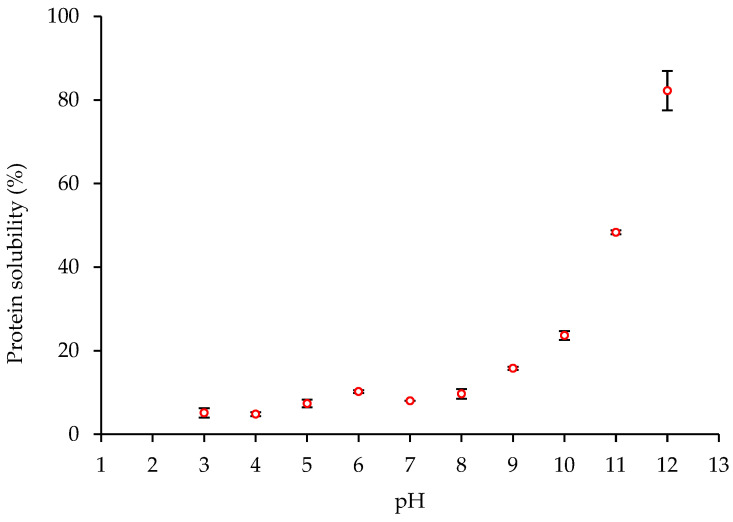
Protein solubility/pH profile of a commercial hempseed protein isolate (HPI) at different pH values. Each point represents the mean ± standard deviation (SD) of triplicate measurements.

**Figure 2 molecules-28-00915-f002:**
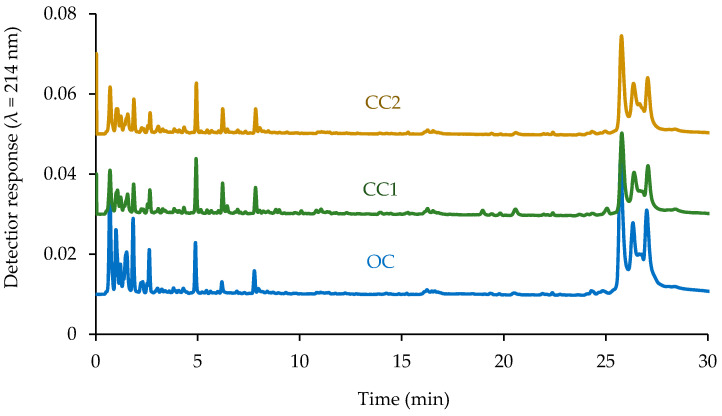
Reverse-phase ultrahigh-performance liquid chromatography (RP-UPLC) profiles of protein isolates from an organically (OC) and two conventionally (CC1 and CC2) cultivated hempseeds.

**Figure 3 molecules-28-00915-f003:**
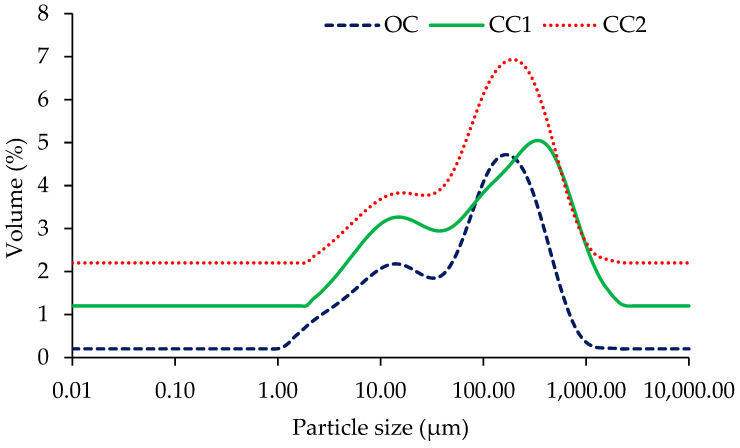
Particle size distribution of 5% (*w*/*v*, on a protein basis) aqueous reconstituted protein extracts from one organically (OC) and two conventionally (CC1 and CC2) cultivated hempseeds.

**Figure 4 molecules-28-00915-f004:**
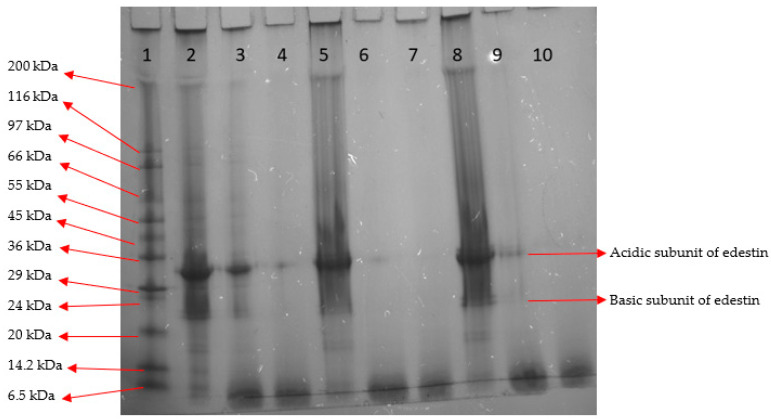
Sodium dodecyl sulfate polyacrylamide gel electrophoresis (SDS-PAGE) profile of 1, molecular mass marker; 2, 5, 8, intact; 3, 6, 9, gastric digested; and 4, 7 and 10, gastrointestinal digested protein samples from an organically (OC) and two conventionally (CC1 and CC2) cultivated hempseeds, respectively.

**Figure 5 molecules-28-00915-f005:**
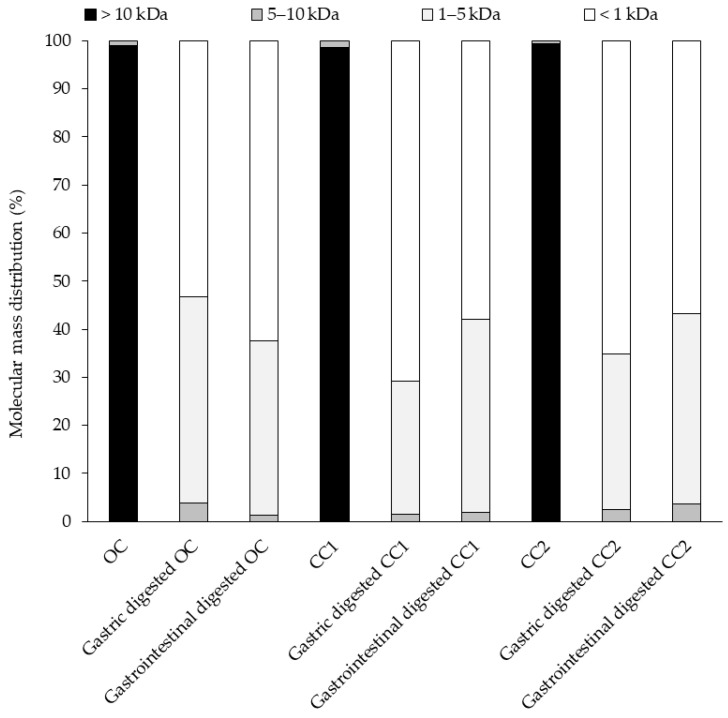
Molecular mass distribution profiles of protein extracts from one organically (OC) and two conventionally (CC1 and CC2) cultivated hempseeds and their associated gastric and gastrointestinal digested samples.

**Table 1 molecules-28-00915-t001:** Proximate analysis of one organically cultivated (OC) and two conventionally (CC1 and CC2) Finola variety hempseeds and a commercial hempseed protein isolate (HPI).

Parameters (%, *w*/*w*)	OC	CC1	CC2	HPI
Moisture	7.30 ± 0.24 ^a^	6.37 ± 0.21 ^b^	6.37 ± 26 ^b^	4.45 ± 0.05 ^c^
Lipid	24.71 ± 1.45 ^a^	26.98 ± 0.72 ^a^	31.32 ± 1.56 ^b^	10.13 ± 0.36 ^c^
Ash	4.92 ± 0.11 ^a^	4.88 ± 0.17 ^a^	5.05 ± 0.05 ^a^	8.38 ± 0.16 ^b^
Protein	23.73 ± 1.38 ^a^	21.93 ± 0.14 ^a^	21.18 ± 0.39 ^a^	41.20 ± 1.25 ^b^
Dietary fiber	23.0 ± 1.3 ^a^	26.8 ± 0.6 ^b^	26.5 ± 0.6 ^b^	18 *
Non-protein nitrogen	2.61 ± 0.19 ^a^	3.28 ± 0.07 ^b^	2.84 ± 0.07 ^a^	3.16 ± 0.14 ^b^

* Value reported by the manufacturer; different letters in each row represent significant difference (*p* < 0.05); n = 3 for all parameters except dietary fiber with n = 2.

**Table 2 molecules-28-00915-t002:** Specific surface area (SSA) and particle size distribution of 5% (*w*/*v*, on a protein basis) aqueous reconstituted protein extracts from one organically (OC) and two conventionally (CC1 and CC2) cultivated hempseed samples.

Parameters	OC	CC1	CC2
SSA (m^2^/g)	0.3 ± 0.0 ^a^	0.2 ± 0.0 ^b^	0.2 ± 0.1 ^ab^
Uniformity	1.2 ± 0.1 ^a^	1.6 ± 0.4 ^a^	1.2 ± 0.4 ^a^
D[4, 3] (µm)	137.0 ± 9.3 ^a^	228.4 ± 29.9 ^b^	176.96 ± 27.8 ^ab^
D[3, 2] (µm)	17.5 ± 0.5 ^a^	26.4 ± 3.8 ^b^	32.3 ± 11.1 ^b^
d(0.1) (µm)	6.0 ± 0.2 ^a^	8.7 ± 1.2 ^b^	12.0 ± 5.7 ^ab^
d(0.5) (µm)	89.0 ± 0.9 ^a^	120.0 ± 23.0 ^b^	115.7 ± 32.1 ^ab^
d(0.9) (µm)	340.3 ± 33.8 ^a^	598.7 ± 103.8 ^b^	423.4 ± 69.5 ^ab^
d(0.9) − d(0.1) (µm)	334.3	590.0	411.4

* Different letters in each row show significant difference (*p* < 0.05), n = 3.

**Table 3 molecules-28-00915-t003:** Degree of hydrolysis (DH%), 2,2′-azino-bis(3-ethylbenzothiazoline-6-sulfonic acid) radical (ABTS^●^) scavenging activity and lipase inhibitory/activatory activity of protein extracts from one organically (OC) and two conventionally (CC1 and CC2) cultivated hempseeds prior to and after *in vitro* simulated gastric and gastric followed by intestinal digestion.

Sample	DH (%)	ABTS^●^ Scavenging Activity (%) at 1 mg/mL (Protein Basis)	Lipase Inhibition/Activation (%) at 1 mg/mL (Protein Basis)
OC		16.9 ± 0.6 ^a^	22.0 ± 15.6 ^a^ (inhibition)
Gastric digested	5.9 ± 1.1 ^a^	86.9 ± 1.3 ^d^	21.2 ± 11.5 ^a^ (inhibition)
Gastrointestinal digested	8.1 ± 0.9 ^b^	85.9 ± 0.4 ^d^	21.7 ± 11.9 ^a^ (inhibition)
CC1		13.7 ± 0.8 ^b^	−36.2 ± 16.6 ^b^ (activation)
Gastric digested	7.5 ± 1.9 ^ab^	82.4 ± 0.6 ^c^	−41.2 ± 21.1 ^b^ (activation)
Gastrointestinal digested	11.0 ± 1.5 ^c^	83.3 ± 0.3 ^c^	−40.9 ± 17.3 ^b^ (activation)
CC2		14.0 ± 0.6 ^b^	−42.2 ± 18.1 ^b^ (activation)
Gastric digested	5.6 ± 0.6 ^a^	83.8 ± 0.6 ^c^	−43.5 ± 20.5 ^b^ (activation)
Gastrointestinal digested	8.7 ± 0.4 ^b^	85.2 ± 0.8 ^cd^	−35.2 ± 18.6 ^b^ (activation)

* Different letters in each column show significant differences (*p* < 0.05), n = 3.

## Data Availability

The data that support the findings of this study are available from the corresponding author upon reasonable request.

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
