# Peer review of "In Vitro* Digestibility, Biological Activity, and Physicochemical Characterization of Proteins Extracted from Conventionally and Organically Cultivated Hempseed (*Cannabis sativa* L.)"

_molecules, 2023, doi:10.3390/molecules28030915_

Round 1
Reviewer 1 Report
The main objectives of Khalesi et al. study were to extract proteins from conventionally and organically cultivated hemp, and to characterize the extracted hempseed proteins in terms of their physicochemical, digestibility, and in vitro bioactive properties. Although the manuscript is attractive, there are some concerns that should be addressed.
-Generally, the manuscript is well organized but there are some typographical and grammatical errors.
-The paper title is well stated, it is informative and concise.
-Abstract is well structured.
L 25-26: "Hemp (Cannabis sativa L.) is ... Cannabinaceae family" should be changed to "Hemp (Cannabis sativa L.) is an annual herbaceous plant that has been widely used due to its industrial (10.3906/bot-1907-15), ornamental (https://doi.org/10.3390/plants11182383), and pharmaceutical (https://doi.org/10.1016/j.biotechadv.2022.108074) applications".
-The introduction was not well written, and it is too briefly presenting the subject and research problem.
-Material and research methods are presented appropriately. The experimental setup and the description in the methods section are well structured, and the statistical analysis is correctly performed.
-The results obtained in this study are interesting. Results are presented correctly.
-In general, the discussion was not well written. This part should be improved.
Author Response
Responses to reviewers comments – Manuscript Molecules-2096702
Reviewer #1
Comment: The main objectives of Khalesi et al. study were to extract proteins from conventionally and organically cultivated hemp, and to characterize the extracted hempseed proteins in terms of their physicochemical, digestibility, and in vitro bioactive properties. Although the manuscript is attractive, there are some concerns that should be addressed.
Response: Thank you for your valuable and comprehensive comments. We have addressed your comments as outlined below.
Comment: Generally, the manuscript is well organized but there are some typographical and grammatical errors.
Response: Thank you. We have now double checked the language and corrected any typos.
Comment: The paper title is well stated, it is informative and concise.
Response: Thank you.
Comment: Abstract is well structured.
Response: Thank you
Comment: L 25-26: "Hemp (Cannabis sativa L.) is ... Cannabinaceae family" should be changed to "Hemp (Cannabis sativa L.) is an annual herbaceous plant that has been widely used due to its industrial (10.3906/bot-1907-15), ornamental (https://doi.org/10.3390/plants11182383), and pharmaceutical (https://doi.org/10.1016/j.biotechadv.2022.108074) applications".
Response: Thank you. We have now added this statement to the Introduction section.
Comment: The introduction was not well written, and it is too briefly presenting the subject and research problem.
Response: Thank you. We have now added a new paragraph explaining the importance of plant proteins and the applications of hemp products/co-products. The Introduction also consists of information on the protein profile, the digestibility, and the bioactivity of hemp proteins. In addition, the gap in knowledge and the objective of the current study has been given in the last paragraph of the Introduction.
Comment: Material and research methods are presented appropriately. The experimental setup and the description in the methods section are well structured, and the statistical analysis is correctly performed.
Response: Thank you very much for your positive feedback.
Comment: The results obtained in this study are interesting. Results are presented correctly.
Response: Thank you.
Comment: In general, the discussion was not well written. This part should be improved.
Response: Thank you. We have included a discussion section after each section of the results. The relevant literature has been cited in each section, the perspective and the conclusions for each section have been provided. We believe that this approach provides a comprehensive exposition of the results in terms of the current state of the art.

Reviewer 2 Report
After reading the publication, I do not make any significant formal or substantive reservations.
However, in the „Sample preparation” (section 3.2), I see no detailed procedure for seed pre-treatment before studying. For example, was it an air-dried material?
Author Response
Responses to reviewers comments – Manuscript Molecules-2096702
Reviewer #2
Comment: After reading the publication, I do not make any significant formal or substantive reservations. However, in the „Sample preparation” (section 3.2), I see no detailed procedure for seed pre-treatment before studying. For example, was it an air-dried material?
Response: Thank you for your positive feedback. The seeds which were supplied by the grower were initially physically cleaned to only take the whole seeds and to remove any extraneous materials. The seeds were de-oiled, as described in Section 3.4, and the protein was extracted from the de-oiled seeds.

Round 2
Reviewer 1 Report
All the comments have been addressed. The current version of the manuscript can be published in Molecules.